# MASLab: A Unified and Comprehensive Codebase for LLM-based Multi-Agent Systems

## Abstract

LLM-based multi-agent systems (MAS) have demonstrated significant potential in enhancing single LLMs to address complex and diverse tasks in practical applications. Despite considerable advancements, the field lacks a unified codebase that consolidates existing methods, resulting in redundant re-implementation efforts, unfair comparisons, and high entry barriers for researchers. To address these challenges, we introduce MASLab, a unified, comprehensive, and research-friendly codebase for LLM-based MAS. (1) MASLab integrates over 20 established methods across multiple domains, each rigorously validated by comparing step-by-step outputs with its official implementation. (2) MASLab provides a unified environment with various benchmarks for fair comparisons among methods, ensuring consistent inputs and standardized evaluation protocols. (3) MASLab implements methods within a shared streamlined structure, lowering the barriers for understanding and extension. Building on MASLab, we conduct extensive experiments covering 10+ benchmarks and 8 models, offering researchers a clear and comprehensive view of the current landscape of MAS methods. MASLab will continue to evolve, tracking the latest developments in the field, and invite contributions from the broader open-source community.

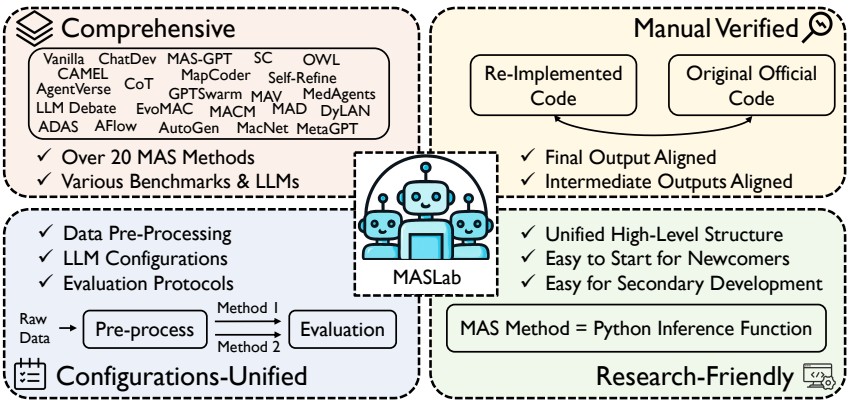

Figure 1: MASLab: A unified, comprehensive, and research-friendly codebase for LLM-based MAS. We support fairly comparing over 20 methods, whose correctness are manually verified.

## 1 Introduction

Large language models (LLMs) (OpenAI, 2023; Anthropic, 2024; Guo et al., 2025; Dubey et al., 2024; Yang et al., 2024b) have seen remarkable success across various domains (Chen et al., 2021a; Park et al., 2023; Tu et al., 2024; Wu et al., 2023). However, individual LLMs face inherent limitations including unreliable generation (Zhou et al., 2024; Wolf et al., 2024), hallucinations (Zhang et al., 2023; Min et al., 2023), and difficulties with complex multi-step tasks (Dziri et al., 2023; Hadi et al., 2023), which hinder their effectiveness in tackling diverse real-world applications.

These limitations have spurred development of LLM-based multi-agent systems (MAS) (Qian et al., 2024; Li et al., 2023; Hu et al., 2025b; Ye et al., 2025), where multiple agents, each with distinct roles, contexts, and tools, collaborate to address complex tasks more effectively. MAS has shown

Table 1: Descriptions of 24 methods that MAS-Lab currently support. We show several critical perspectives of MAS methods. (1) Role: whether agents' roles in the method is fixed or dynamic. (2) Topo.: whether the topology in the method is fixed or dynamic. (3) Tool: whether the method includes tool usage. (4) Optim.: whether the method is optimizable. (5) Generalization: whether the method can generalize to handle diverse tasks.

| No. | Methodology | Venue | Role | Topo. | Tool | Optim. | Generalization |
|---|---|---|---|---|---|---|---|
| | Single-Agent Baselines | | | | | | |
| ① | Vanilla LLM | - | Fixed | Fixed | No | No | Yes |
| ② | CoT (Wei et al., 2022) | NeurIPS 2022 | Fixed | Fixed | No | No | Yes |
| | Multi-Agent Systems for General Tasks | | | | | | |
| ③ | CAMEL (Li et al., 2023) | NeurIPS 2023 | Fixed | Fixed | No | No | Yes |
| ④ | AutoGen (Wu et al., 2024) | ICLR-W 2024 | Fixed | Fixed | Yes | No | Yes |
| ⑤ | Self-Consistency (Wang et al., 2024b) | ICLR 2024 | Fixed | Fixed | No | No | Yes |
| ⑥ | AgentVerse (Chen et al., 2024) | ICLR 2024 | Dynamic | Fixed | No | No | Yes |
| ⑦ | LLM Debate (Du et al., 2024) | ICML 2024 | Fixed | Fixed | No | No | Pre-defined Roles |
| ⑧ | GPTSwarm (Zhuge et al., 2024) | ICML 2024 | Fixed | Dynamic | Yes | Yes | Validation-Required |
| ⑨ | DyLAN (Liu et al., 2024) | COLM 2024 | Fixed | Dynamic | No | No | Pre-defined Roles |
| ⑩ | MAD (Liang et al., 2024) | EMNLP 2024 | Fixed | Fixed | No | No | Pre-defined Roles |
| ⑪ | Self-Refine (Madaan et al., 2024) | NeurIPS 2024 | Fixed | Fixed | No | No | Yes |
| ⑫ | MacNet (Qian et al., 2025) | ICLR 2025 | Fixed | Fixed | No | No | Pre-defined Roles |
| ⑬ | ADAS (Hu et al., 2025b) | ICLR 2025 | Fixed | Fixed | Yes | Yes | Validation-Required |
| ⑭ | AFlow (Zhang et al., 2025b) | ICLR 2025 | Fixed | Fixed | Yes | Yes | Validation-Required |
| ⑮ | MAV (Lifshitz et al., 2025) | ICLR-W 2025 | Fixed | Fixed | No | No | Yes |
| ⑯ | MAS-GPT (Ye et al., 2025) | ICML 2025 | Dynamic | Dynamic | Yes | Yes | Yes |
| | Multi-Agent Systems for Coding Tasks | | | | | | |
| ⑰ | MetaGPT (Hong et al., 2024) | ICLR 2024 | Fixed | Fixed | Yes | No | Coding-Specific |
| ⑱ | ChatDev (Qian et al., 2024) | ACL 2024 | Fixed | Fixed | Yes | No | Coding-Specific |
| ⑲ | MapCoder (Islam et al., 2024) | ACL 2024 | Fixed | Fixed | Yes | No | Coding-Specific |
| ⑳ | EvoMAC (Hu et al., 2025c) | ICLR 2025 | Dynamic | Dynamic | Yes | No | Coding-Specific |
| | Multi-Agent Systems for Mathematical Tasks | | | | | | |
| ㉑ | MACM (Lei et al., 2024) | NeurIPS 2024 | Fixed | Fixed | No | No | Math-Specific |
| | Multi-Agent Systems for Scientific Tasks | | | | | | |
| ㉒ | MedAgents (Tang et al., 2024b) | ACL-F 2024 | Fixed | Fixed | No | No | Medicine-Specific |
| | Multi-Agent Systems for Tool-Required Tasks | | | | | | |
| ㉓ | OWL-Roleplaying (Hu et al., 2025a) | GitHub 2025 | Fixed | Fixed | Yes | No | Yes (with Proper Tools) |
| ㉔ | ReAct-MASLab (Yao et al., 2023) | ICLR 2023 | Fixed | Fixed | Yes | No | Yes (with Proper Tools) |

promise in diverse applications including code generation (Qian et al., 2024; Hong et al., 2024), mathematical reasoning (Lei et al., 2024; Imani et al., 2023), academic research (Lu et al., 2024; Schmidgall et al., 2025), and data synthesis (Pang et al., 2024; Tang et al., 2024a). The field has rapidly evolved from fixed, manually-designed systems (Qian et al., 2024; Du et al., 2024; Hong et al., 2024; Liang et al., 2024; Chen et al., 2024; Lei et al., 2024) to dynamic systems with adaptable agent roles (Hu et al., 2025b; Ye et al., 2025; Zhang et al., 2025b; Liu et al., 2024; Zhuge et al., 2024). This ongoing evolution is steering the field towards greater automation and generalization, with the potential to create more intelligent systems.

Despite the rapid progress in LLM-based MAS, the field lacks a unified codebase that consolidates the various methods and algorithms. This gap results in several critical issues that hinder the field's long-term advancement: **(1) Redundant effort.** Without shared, accessible resources, researchers expend significant time reimplementing existing works, diverting effort from innovative contributions. **(2) Unfair comparison.** Varied implementation designs of individual codebases, such as differing dataset preprocessing and evaluation protocols, complicate fair and reliable comparisons across methods. **(3) High entry barriers.** Newcomers face difficulties navigating through disparate repositories, with no clear starting points. Addressing these challenges is crucial to accelerate research and promote cohesive progress in the field. However, unifying massive methods—that originally employ distinct codebase styles, architectures, and dependencies—into one codebase poses significant challenges. This requires not only substantial efforts for re-implementation and verification, but also a comprehensive understanding of all methods to enable unification.

To bridge this gap, we present **MASLab**, the first unified codebase for LLM-based MAS, integrating over 20 established methods with a coherent structure and standardized evaluations; see overview in

Table 1. **(1)** MASLab consolidates diverse research across domains including general tasks (Chen et al., 2024), coding (Qian et al., 2024), and mathematics (Lei et al., 2024)—covering representative advancements from March 2023 through March 2025. Each method integrated into MASLab has been rigorously verified by comparing step-by-step outputs with its official implementation, greatly reducing redundant reimplementation efforts for future researchers. **(2)** MASLab supports unified evaluations across a wide array of benchmarks, ensuring consistent inputs and standardized evaluation protocols. This facilitates reliable and fair comparisons, emphasizing core methodological differences rather than implementation disparities. **(3)** All methods are implemented within a streamlined, high-level structure, where each is encapsulated as a core inference function that processes a query and delivers the MAS response. This transparent structure explicitly highlights key methodological components, significantly lowering entry barriers and enabling researchers to easily understand, extend, and innovate upon existing approaches.

Based on MASLab, we conduct comprehensive experiments to benchmark the implemented methods, offering the research community a clear understanding of the current landscape of LLM-based MAS. Our evaluation spans 10+ benchmarks spanning diverse domains—including general question answering, mathematics, coding, science, and medicine—using 8 LLM backbones including Llama-3.3-70B-Instruct, Qwen-2.5-7/14/32/72B-Instruct, and GPT-4o-mini/4.1-mini/4.1 models. Our analysis examines the impact of varying evaluation protocols adopted by prior studies, the scaling behavior with respect to method configuration and model size, and failure cases. Notably, we demonstrate that discrepancies in evaluation protocols can lead to substantial variation in performance rankings, underscoring the importance of a unified codebase for fair and reproducible comparisons.

## 2  RELATED WORK

**LLM-based MAS.** LLM-based multi-agent systems (MAS) extend the capabilities of LLMs by enabling collaborative interactions among multiple agents. CAMEL (Li et al., 2023) and AutoGen (Wu et al., 2024) primarily focus on two-agent (user–assistant) role-playing, while MetaGPT (Hong et al., 2024) and ChatDev (Qian et al., 2024) assign multiple specialized roles (e.g., coder, reviewer) for fixed software development pipeline. Debate-style systems (Du et al., 2024; Liang et al., 2024) employ multiple agents to propose and criticize solutions. AgentVerse (Chen et al., 2024) and DyLAN (Liu et al., 2024) allow iterative adjustment of team configurations during task execution.

While these fixed-role architectures demonstrate the potential of MAS, they rely heavily on manually defined roles and workflows, limiting generalizability across tasks. To address this, recent works explore automatic workflow generation (Ye et al., 2025; Hu et al., 2025b; Zhang et al., 2024; 2025a;c). GPTSwarm (Zhuge et al., 2024) models agents as an optimizable graph of LLM operations refined via validation feedback. Similarly, ADAS (Hu et al., 2025b) and AFlow (Zhang et al., 2025b) leverage a strong meta-agent to iteratively design agentic workflows. MAS-GPT (Ye et al., 2025) trains an LLM that generates an executable MAS based on each user query.

However, these methods are implemented in isolated codebases, leading to redundant efforts, inconsistent evaluations, and steep entry barriers. MASLab resolves these issues by providing a unified and comprehensive codebase that supports all of the above methods within an extensible framework.

**LLM-agent codebase.** In parallel with algorithmic advances, several open-source frameworks have emerged to facilitate the development of LLM-based agents. CAMEL (Li et al., 2023) and AutoGen (Wu et al., 2024) introduce conversational agent frameworks based on role-playing. LangChain (LangChain, 2025), LangGraph (LangGraph, 2025), and OpenAgents (Xie et al., 2024) provide low-code environments for constructing LLM-driven applications and workflows. However, none of these frameworks are designed specifically for research purposes: they lack implementations of representative multi-agent methods from the existing literature and offer limited support for systematic evaluation. In contrast, our MASLab offers the first all-in-one research-friendly codebase that integrates the community's collective progress in LLM-based MAS.

## 3  MASLAB

MASLab is a unified, comprehensive, research-oriented codebase for LLM-based multi-agent systems (MAS). It consolidates over 20 published MAS methods with consistent inference basic configurations and unified evaluation protocols, facilitating researchers for fast and fair algorithmic comparisons. All methods are verified by comparing their intermediate outputs with the official implementations.

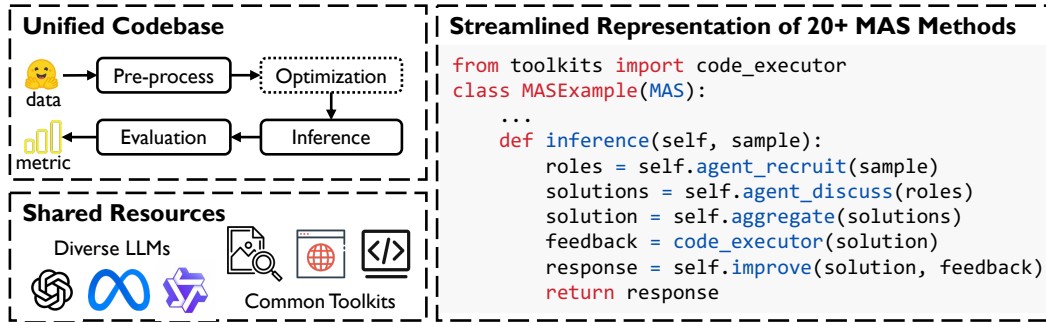

Figure 2: Overview of MASLab codebase. MASLab incorporates and unifies the whole pipeline from data pre-processing to evaluation, ensuring that inputs to all methods are aligned, non-algorithmic configurations are standardized, and the evaluation protocols are consistent and accurate. All 20+ methods are represented by a similar streamlined structure of python class.

## 3.1 INFERENCE OF MAS

In order to unify and streamline the diverse MAS codebases in the field, MASLab focuses on four key aspects during inference that ensure consistency and fairness across different methods: representation of MAS, inputs, configurations, and accessible resources. These aspects are designed to eliminate the disparities that have traditionally hindered cross-method comparisons and replication of results.

**Streamlined representation of MAS.** Each MAS method within MASLab is abstracted into a Python class, all of which inherit from a common base class. This base class provides shared functionality across methods, such as making LLM requests and tracking LLM token consumptions. The core of each method is the *inference* function, which takes a data sample (e.g., a mathematical problem) as input and outputs the solution generated by the MAS. By standardizing the representation in this manner, the structure of each MAS approach is simplified, allowing researchers to gain a clear understanding of the key steps involved by merely inspecting the inference function. In many cases, the inference process is further modularized, with specific components encapsulated as additional functions to highlight the different stages of task-solving, such as team recruitment and code execution. This design ensures that the complexity inherent in different MAS methods is handled in a consistent, easily interpretable manner, while preserving the unique features of each individual approach. For optimization-based methods (Hu et al., 2025b; Zhuge et al., 2024; Zhang et al., 2025b), another core *optimization* function will process a validation set to produce an optimized MAS. See re-implementation notes in Section E.

**Consistent inputs.** MASLab standardizes input preprocessing for all MAS methods, ensuring fair comparisons by eliminating discrepancies. For instance, prior implementations of MapCoder (Islam et al., 2024), Reflexion (Shinn et al., 2023), and EvoMAC (Hu et al., 2025c) use different preprocessing on the MBPP dataset, making performance differences hard to interpret. MASLab's unified preprocessing pipeline ensures that all methods operate on identical data, relieving researchers of the need to manually prepare datasets.

**Shared resources.** MASLab unifies the underlying resources required by MAS methods, including LLMs and external tools. It supports both externally hosted APIs and locally deployed models, covering a wide range of widely used LLMs. The integrated toolkit provides common utilities such as code execution (secured via sandboxing (Bytedance, 2025)), web search, and image analysis—capabilities frequently required across MAS designs. These shared components eliminate redundant engineering effort and facilitate reproducibility. Moreover, MASLab is designed to be extensible and compatible with ongoing open-source developments (e.g., MCP (Anthropic, 2025)), ensuring long-term adaptability.

**Unified configurations.** MASLab standardizes non-algorithmic configurations across all methods to ensure fair and consistent comparisons. This includes aligning LLM settings (e.g., maximum token limits) and tool parameters (e.g., timeout durations for code execution). Such uniformity

eliminates confounding factors introduced by implementation-level differences, allowing performance comparisons to reflect true methodological distinctions.

## 3.2 EVALUATION OF MAS

Accurate, automated, and scalable evaluation protocols are always essential for all AI fields. However, existing MAS works often adopt inconsistent evaluation procedures, introducing confounding variables that hinder fair comparison. For example, certain methods may be tailored to specific evaluation heuristics (e.g., rule-based string matching) which can be gamed by emphasizing format-specific prompts for agents, thereby inflating performance without reflecting true intelligence gains. These issues underscore the need for standardized, robust evaluation protocols that reflect genuine task-solving capabilities rather than formatting tricks.

**Evaluating responses with ground-truth answers.** To address this, MASLab adopts a unified evaluation framework centered on LLM-based evaluation methods grounded in ground-truth answers, designed to assess semantic correctness rather than superficial formatting. We support two primary variants: (1) A two-step pipeline using general-purpose LLMs, which first extracts a final answer from the MAS-generated output based on the query, and then compares it against the ground-truth to determine correctness; (2) A direct scoring approach using task-specific evaluators (e.g., xVerify (Chen et al., 2025)), which are fine-tuned to assess correctness across various domains. In addition, MASLab includes three commonly used rule-based evaluation strategies from the MAS literature.

Surprisingly, our empirical results on MATH (Hendrycks et al., 2021) benchmark (Figure 3) show that evaluation protocol choice significantly affects both absolute scores and method rankings. For instance, under the LLM-based two-step evaluation, MAV (Lifshitz et al., 2025) ranks 1st, but drops to 10th under DyLAN's rule-based scheme (Liu et al., 2024). Conversely, DyLAN itself rises from 5th to 3rd. Similarly, AgentVerse's accuracy drops from 79.0 to 25.6 when switching from the LLM-based two-step evaluation to the Hendrycks-style rule-based metric (Hendrycks et al.,

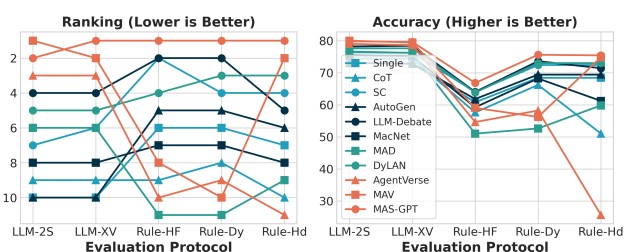

Figure 3: Evaluation (5 different protocols) of methods using Llama-3.3-70B-Instruct as the backend on MATH. The rankings of methods could be significantly different under different evaluation protocols, emphasizing the need for accurate and unified evaluation protocols.

2021). Manual inspection (Table 4) confirms the higher reliability of LLM-based evaluations, with both the two-step and xVerify approaches achieving over 98% agreement with human judgments, whereas the best-performing rule-based method reaches only 65%. Considering performance-cost trade-off, MASLab defaults to using xVerify, while remaining open to improvements as evaluation methodologies evolve.

**Evaluating coding tasks with test cases.** For coding tasks, where ground-truth labels are often unavailable, MASLab similarly promotes LLM-assisted evaluation. Since tools like xVerify are inapplicable in this setting, we employ a two-step approach: (1) an LLM extracts executable code from the MAS output given the original query, and (2) the extracted code is executed against the provided set of test cases to determine correctness. This process ensures that evaluation focuses on functional validity and abstracts away from inconsistencies in format or verbosity. All executions are sandboxed (Bytedance, 2025) to guarantee safety and consistency.

## 4 EMPIRICAL STUDY

**Experimental setups.** Our experiments cover Llama (Llama-3.3-70B-Instruct (Dubey et al., 2024)), Qwen (Qwen-2.5-7/14/32/72B-Instruct (Yang et al., 2024a)), and GPT (GPT-4omini/4.1mini/4.1 (OpenAI, 2024b;a; 2025)) LLMs. We set the max token as 2048 with a temperature of 0.5. Our datasets cover domains including mathematics (MATH (Hendrycks et al., 2021), GSM-Hard (Gao et al., 2023),

Table 2: Results of general methods on diverse domains ( mathematics , science , knowledge , medicine , coding ). Avg-V and Avg-R denotes averaged accuracy value (↑) and rank (↓) across benchmarks. **Best** and second-best numbers are highlighted.

| Method | MATH | GSM-H | AQUA | AIME | SciBe | GPQA | MMLUP | MedMC | HEval | MBPP | Avg-V | Avg-R |
|---|---|---|---|---|---|---|---|---|---|---|---|---|
| Llama-3.3-70B-Instruct | | | | | | | | | | | | |
| Single (Dubey et al., 2024) | 72.8 | 52.8 | 76.0 | 23.3 | 25.5 | 48.0 | 66.8 | 70.4 | 85.4 | 67.9 | 58.9 | 8.1 ± 2.5 |
| CoT (Wei et al., 2022) | 74.4 | 57.0 | 76.8 | 26.7 | 24.7 | 53.0 | 69.8 | 73.8 | 85.4 | 69.7 | 61.1 | 5.5 ± 2.8 |
| SC (Wang et al., 2024b) | 76.2 | 53.4 | 80.3 | 30.0 | 27.9 | 52.5 | 70.6 | 72.6 | 82.3 | 69.7 | 61.6 | 4.8 ± 2.8 |
| AutoGen (Wu et al., 2024) | 72.8 | 53.0 | 79.5 | 20.0 | 21.9 | 41.9 | 66.0 | 69.2 | 51.2 | 62.9 | 53.8 | 9.9 ± 2.5 |
| Debate (Du et al., 2024) | 78.4 | 53.6 | 80.3 | 30.0 | 27.9 | 51.0 | **73.6** | **75.0** | 84.8 | 69.7 | 62.4 | 3.5 ± 1.9 |
| MAD (Liang et al., 2024) | 76.2 | 52.6 | 78.3 | **33.3** | 23.7 | 50.0 | 69.8 | 71.0 | 75.0 | 56.9 | 58.7 | 8.1 ± 3.1 |
| AgentVerse (Chen et al., 2024) | 78.6 | 51.2 | 79.5 | 23.3 | 25.7 | 51.0 | 70.4 | 72.6 | **87.8** | **71.3** | 61.2 | 4.4 ± 3.2 |
| DyLAN (Liu et al., 2024) | 77.6 | 53.6 | 78.3 | **33.3** | 26.7 | **54.0** | 70.4 | 72.6 | 82.9 | 70.1 | 62.0 | 4.7 ± 2.8 |
| MacNet (Qian et al., 2025) | 75.2 | 56.6 | 77.2 | 26.7 | 23.5 | 51.0 | 64.0 | 69.8 | 86.6 | 67.1 | 59.8 | 7.2 ± 3.0 |
| MAV (Lifshitz et al., 2025) | 79.4 | 35.6 | 65.8 | 30.0 | 24.3 | 45.0 | 61.0 | 68.6 | 78.0 | 69.1 | 55.7 | 9.3 ± 3.0 |
| AFlow-Math (Zhang et al., 2025b) | **82.2** | 59.8 | 77.2 | 26.7 | 25.3 | 48.0 | 68.6 | 69.2 | 84.2 | 69.7 | 61.1 | 5.9 ± 3.0 |
| MAS-GPT (Ye et al., 2025) | 79.8 | **67.0** | **80.7** | **33.3** | 26.9 | 48.5 | 71.2 | 72.0 | 86.6 | 70.3 | **63.6** | **3.3 ± 2.6** |
| Qwen-2.5-72B-Instruct | | | | | | | | | | | | |
| Single (Yang et al., 2024a) | 82.4 | 63.2 | 79.5 | 20.0 | 28.1 | 45.0 | 69.8 | 67.6 | 88.4 | 76.5 | 62.1 | 6.2 ± 2.4 |
| CoT (Wei et al., 2022) | 83.0 | 64.2 | 80.3 | 16.7 | 26.3 | 47.0 | 71.2 | 67.8 | **93.9** | 75.8 | 62.6 | 5.1 ± 2.1 |
| SC (Wang et al., 2024b) | 86.0 | 63.2 | **83.5** | 20.0 | 28.3 | 49.0 | 73.0 | 69.0 | 90.2 | 75.8 | **63.8** | **2.8 ± 1.8** |
| AutoGen (Wu et al., 2024) | 81.4 | 63.2 | 78.3 | 13.3 | 26.5 | 44.4 | 70.2 | 68.4 | 75.6 | 54.9 | 57.6 | 8.8 ± 2.6 |
| Debate (Du et al., 2024) | 85.4 | 62.0 | 80.3 | 20.0 | 26.9 | 49.0 | **74.0** | 71.0 | 90.2 | 77.6 | 63.6 | 3.2 ± 2.2 |
| MAD (Liang et al., 2024) | 83.8 | 61.6 | 80.3 | 20.0 | 26.9 | 47.0 | 65.6 | 67.4 | 72.0 | 66.3 | 59.1 | 7.6 ± 3.4 |
| AgentVerse (Chen et al., 2024) | 82.8 | 57.6 | 79.5 | 13.3 | 25.9 | 46.0 | 72.0 | **71.2** | 86.6 | 77.6 | 61.2 | 6.4 ± 3.1 |
| DyLAN (Liu et al., 2024) | 84.2 | 62.4 | 82.3 | 20.0 | 24.7 | 43.4 | 71.2 | 70.0 | 79.9 | 76.8 | 61.5 | 6.1 ± 3.2 |
| MacNet (Qian et al., 2025) | 82.2 | 63.0 | 80.3 | 10.0 | 25.1 | 42.4 | 65.4 | 63.8 | 87.2 | 75.5 | 59.5 | 8.5 ± 3.3 |
| MAV (Lifshitz et al., 2025) | 82.2 | 20.4 | 48.0 | 0.0 | 14.5 | 46.5 | 61.2 | 65.4 | 76.2 | 74.0 | 48.8 | 9.9 ± 3.2 |
| AFlow-Math (Zhang et al., 2025b) | 84.8 | **68.4** | 78.7 | **23.3** | 28.3 | 47.5 | 69.2 | 66.2 | 87.8 | 75.5 | 63.0 | 5.7 ± 3.5 |
| MAS-GPT (Ye et al., 2025) | **87.0** | 65.4 | 78.3 | 20.0 | 28.1 | 49.0 | 72.6 | 66.2 | 89.0 | **78.0** | 63.4 | 4.0 ± 3.2 |

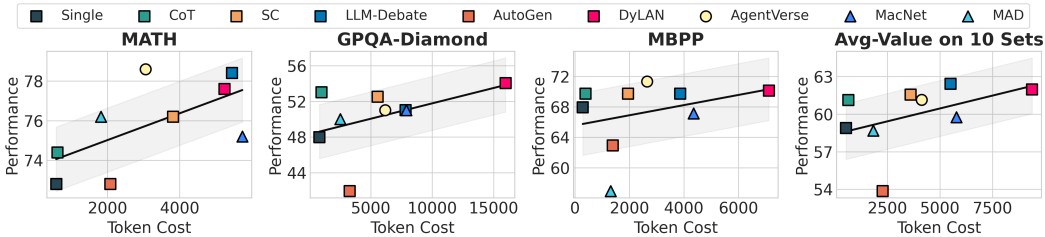

Figure 4: Trade-off between performance and cost. For fair comparisons, we only plot methods that do not involve tool usage. Methods above the fitted line are more cost-effective.

AQUA-RAT (Ling et al., 2017), AIME-2024), science (SciBench (Wang et al., 2024a), GPQA (Rein et al., 2023)), knowledge (MMLU-Pro (Wang et al., 2024c)), medicine (MedMCQA (Pal et al., 2022)), coding (HumanEval (Chen et al., 2021b), MBPP (Austin et al., 2021)), and AI-assistant (GAIA (Mialon et al., 2024)).

## 4.1 CURRENT LANDSCAPE OF MAS METHODS

**Comparisons of general MAS on diverse domains.** able 2 compares general MAS methods across diverse domains (mathematics, science, knowledge, medicine, coding), revealing that: (1) no method rules on all domains, suggesting that there is large room for future methods that could generalize well on more domains. (2) Backend models significantly affect performance; e.g., AgentVerse (Chen et al., 2024) and DyLAN (Liu et al., 2024) achieve better performance than Single using Llama-3.3-70B-Instruct while worse using Qwen-2.5-72B-Instruct. One hypothesis is that Llama-3.3-70B-Instruct has better collaboration capability than Qwen-2.5-72B-Instruct as we see that the gap between best-performing MAS and Single reduce from 4.7% to 1.7%. This suggests an interesting future direction of exploring the most suitable LLM for MAS or training more appropriate ones. (3) MAS-GPT (Ye et al., 2025) and LLM-Debate (Du et al., 2024) perform best overall, owing to their dataset-agnostic designs. Beyond accuracy, we analyze performance-cost trade-offs in Figure 4 and Figure 11. Higher accuracy typically requires more tokens, with methods above the fitted line being more cost-effective.

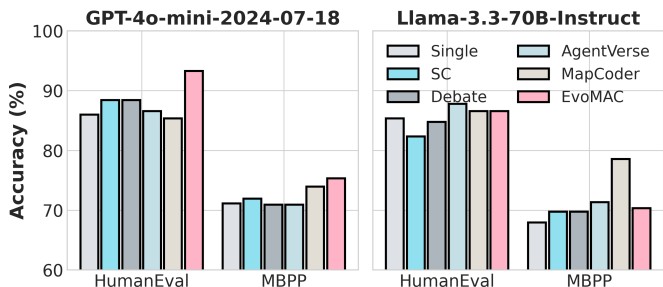
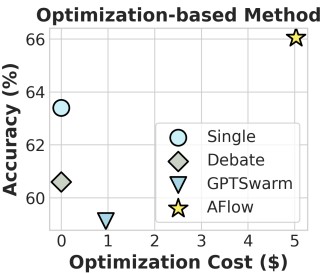

Figure 5: Examining coding-specific methods (MapCoder and EvoMAC). Using GPT-4o-mini, EvoMAC performs best; with Llama-3.3-70B-Instruct, MapCoder leads.

Figure 6: Optimization-based methods (GPTSwarm and AFlow) on MATH.

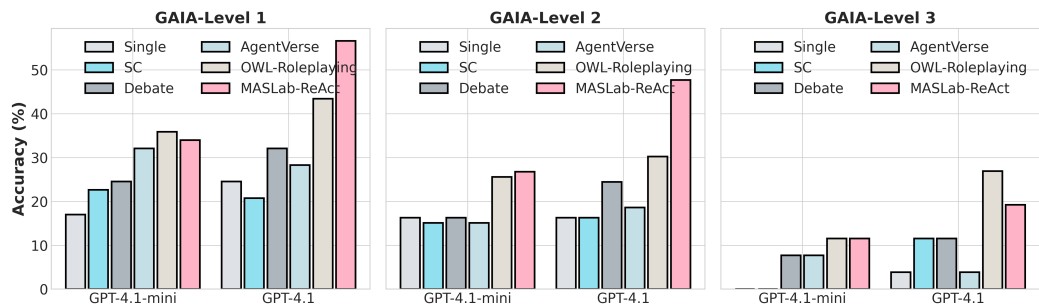

Figure 7: Examining MAS as AI assistants on GAIA (Mialon et al., 2024). (1) Equipping agents with tools (OWL-Roleplaying and MASLab-ReAct) improves MAS performance. (2) The performance gains are more significant using stronger LLMs. (3) Our MASLab-ReAct performs the best.

**Examining coding-specific methods.** We compare two coding-specific methods, MapCoder (Islam et al., 2024) and EvoMAC (Hu et al., 2025c), on HumanEval and MBPP, using GPT-4o-mini and LLama-3.3-70B-Instruct as backends. Results in Figure 5 show that the performance is closely tied to the underlying LLM. Specifically, EvoMAC (Hu et al., 2025c) consistently outperforms others when paired with GPT-4o-mini, whereas MapCoder (Islam et al., 2024) achieves the best results with LLaMA-3.3-70B-Instruct, especially on MBPP. This discrepancy may be attributed to backend-specific prompt optimization: e.g., EvoMAC (Hu et al., 2025c) was primarily developed and tuned on GPT-4o-mini in its original work.

**Examining optimization-based methods.** We compare two optimization-based methods, AFlow (Zhang et al., 2025b) and GPTSwarm (Zhuge et al., 2024), on the MATH (Hendrycks et al., 2021) dataset. Following AFlow's (Zhang et al., 2025b) original setup, we apply Claude-3.5-Sonnet (Anthropic, 2024) as the optimizer and GPT-4o-mini (OpenAI, 2024b) as the executor. The evaluation protocol during testing matches that used in the optimization process (AFlow's rule-based evaluation). Figure 6 reports the required cost for optimization and the achieved performance. We see that AFlow (Zhang et al., 2025b) incurs the most optimization cost while also achieving the best performance; while GPTSwarm (Zhuge et al., 2024) experiences performance drop after optimization in this setup. This discrepancy likely stems from AFlow's LLM-based optimization being more effective than GPTSwarm's numerical approach, suggesting that the strategy of optimization should be carefully considered to ensure effectiveness.

**Examining MAS as AI assistants.** While our earlier experiments primarily focus on standard LLM benchmarks—where improvements from MAS may sometimes appear marginal—this is due to the current lack of benchmarks specifically tailored to MAS. Nevertheless, such evaluations help establish a broad understanding of MAS performance across diverse scenarios.

Here, we evaluate MAS on a more suitable benchmark: GAIA (Mialon et al., 2024), which is designed to assess tool-augmented AI assistants. In this experiment, we provide agents with a suite of tools including a code executor, web search engine, document reader, and image/audio/video analysis utilities (see details in Section D.2). We consider two representative MAS methods with iterative

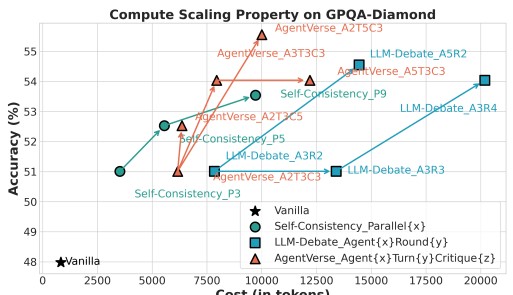
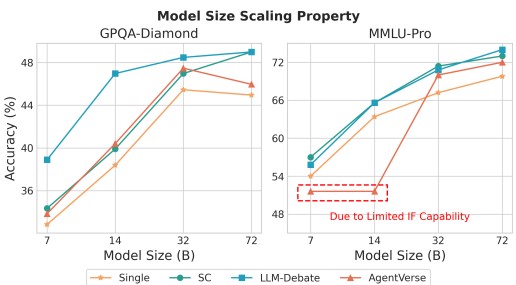

Figure 8: Examining compute-scaling properties on GPQA-Diamond. MASLab offers a platform for readily examining and choosing methods. Here, we see Self-Consistency and AgentVerse achieve better cost-performance trade-off.

Figure 9: Examining size-scaling proprieties on GPQA-Diamond and MMLU-Pro. LLM-Debate performs the best overall. Some methods (e.g., AgentVerse) require the model to attain sufficient capability before MAS can be effective.

planning and action: OWL-Roleplaying (Hu et al., 2025a), and our implemented MASLab-ReAct, inspired by the ReAct paradigm (Yao et al., 2023). We run experiments using two recent OpenAI models—GPT-4.1-mini and GPT-4.1 (OpenAI, 2025). As shown in Figure 7, our findings are as follows: (1) Equipping agents with tools significantly improves MAS performance, surpassing both single-agent baselines and tool-less MAS methods. (2) The performance gains from tools are more pronounced when stronger LLM backends are used. For example, MASLab-ReAct achieves a 91% relative improvement over the single-LLM baseline when using GPT-4.1-mini, and an impressive 171% improvement with GPT-4.1. (3) Table 5 presents the performance–cost trade-off. MASLab-ReAct not only achieves the best performance but also consumes less than half the tokens compared to the second-best method, OWL-Roleplaying. We provide a failure analysis in Figure 10.

## 4.2 SCALING PROPERTIES

As a unified codebase, our MASLab offers a platform for researchers and practitioners to readily examine, explore, and choose different methods. For example, we could use MASLab to explore the scaling properties of different methods by simply modifying some of the configurations.

**Scaling compute / inference times.** We compare three configurable methods—Self-Consistency (Wang et al., 2024b), LLM-Debate (Du et al., 2024), and AgentVerse (Chen et al., 2024)—on GPQA-Diamond (Rein et al., 2023) using Llama-3.3-70B-Instruct (Dubey et al., 2024), to examine which method has the best compute-scaling property. The configurable parameters are: the number of parallel solutions (Self-Consistency); the number of debate agents and debate rounds (LLM-Debate); the number of recruiting agents, loop turns, and criticizing rounds (AgentVerse). Figure 8 shows: (1) Self-Consistency and AgentVerse achieve the best performance-cost trade-off as their dots are mostly on the upper left. (2) Scaling the compute can generally enhance the performance of these examined methods. For AgentVerse, increasing the number of loop turns from 3 to 5 brings the most performance improvement. For LLM-Debate, increasing the number of agents is more effective than increasing the number of rounds in this case.

**Scaling backend model size.** We scale model size using the Qwen-2.5 instruct series (7B, 14B, 32B, 72B), comparing three MAS methods and a single-agent baseline on GPQA-Diamond (Rein et al., 2023) and MMLU-Pro (Wang et al., 2024c). As shown in Figure 9, we observe the following: (1) Overall, the performance of all methods improves with increasing model size, suggesting that stronger LLM backends generally benefit both MAS and single-agent approaches. Notably, LLM-Debate achieves particularly strong gains on GPQA-Diamond.

(2) On MMLU-Pro, two outliers emerge: AgentVerse with 7B and 14B backends shows significantly degraded performance compared to other methods. Manual inspection reveals that these smaller models often fail to follow instruction formats correctly, causing outputs to deviate from the expected response schema (see Section 4.3 for details). (3) These observations indicate that MAS methods relying on precise formatting, intermediate reasoning steps, or structured inter-agent communication, such as role assignment, voting, or sequential planning, may require a minimum threshold of language competency from the backend model. Below this threshold, the benefits of MAS design may be

overshadowed by failures in basic task adherence. This highlights an interesting future direction: designing MAS methods that are more robust to backend model limitations, or adapting interaction protocols to better accommodate smaller, less capable LLMs.

## 4.3 FAILURE ANALYSIS

Here, we explore the reasons why MAS methods fail by analyzing the error logs.

**Format errors.** Format error is a common type of failure in many MAS methods, where LLMs fail to produce responses in the required format. A notable example occurs during the recruiting step in AgentVerse (Chen et al., 2024), where LLMs are tasked with outputting a predefined number of agents in a specific format. To investigate this, we analyze an outlier case from Figure 9 using Qwen-2.5-14B-Instruct as the model backend. We classify incorrect outputs into three categories: wrong answers (i.e., the MAS produces an incorrect final answer), format errors (i.e., the MAS fails to produce a final answer due to formatting issues), and others. As

Table 3: Error analysis of Agent-Verse (Chen et al., 2024) using Qwen-2.5-14B-Instruct as the backend. Expect for that answers are wrong, all errors are caused by format errors.

| Dataset | Wrong | Format | Others |
|---------|-------|--------|--------|
| GPQA-D | 79.66% | 22.34% | 0.00% |
| MMLU-Pro | 47.11% | 52.89% | 0.00% |
| MATH | 42.56% | 57.44% | 0.00% |

shown in Table 3, format errors account for a significant portion of failures. Similar issues are observed in other methods like MAD (Liang et al., 2024) and DyLAN (Liu et al., 2024). These findings underscore a critical challenge in LLM-based MAS: success hinges not only on reasoning or task comprehension but also on the model's ability to meet strict formatting requirements. Improving format adherence or relaxing these constraints could significantly enhance system reliability.

**Error analysis in tool-augmented scenario.** We investigate the performance of OWL-Roleplaying on the GAIA benchmark, which encompasses the most diverse components during task-solving, making it an ideal case study for comprehensive failure analysis. Our analysis reveals that, in this context, failure cases account for 66% of all samples. However, only 36.7% of these failures stem from incorrect final answers, while 45.0% are attributed to errors in tool usage. These findings suggest that future research should focus not only on enhancing agents' tool-handling capabilities but also on improving the quality of tools themselves—particularly their stability and efficiency—to create more robust

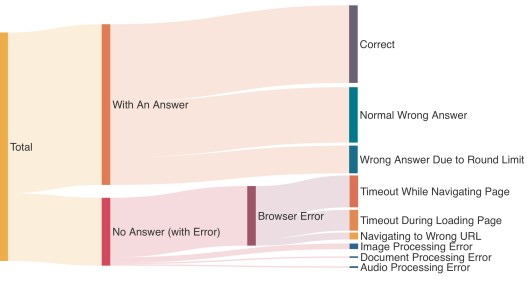

Figure 10: Error analysis of OWL-Roleplaying on GAIA (Mialon et al., 2024) using GPT-4.1 as the model backend.

and effective MAS. We believe that advancements in MCP tools within the open-source community could significantly contribute to the development of MAS.

## 5 CONCLUSIONS

This paper introduces MASLab, a unified, comprehensive, research-friendly codebase for LLM-based multi-agent systems (MAS). (1) MASLab integrates 20+ established methods across multiple domains, each rigorously validated by comparing step-wise outputs with its official implementation. (2) MASLab unifies the whole pipeline from data pre-processing to evaluation, ensuring that all non-algorithmic factors are well aligned for fair comparisons. (3) MASLab implements methods in a shared streamlined structure, lowing entry barriers and simplifying secondary development. Extensive experiments covering 10+ benchmarks and 8 LLMs comprehensively showcase the current landscape of MAS methods. We also provide several analysis, such as exploring the effects of different evaluation protocols in existing works, the compute- and size-scaling properties. Notably, we demonstrate that the discrepancies in evaluation protocols can lead to substantial variation in performance rankings, directly underscoring the importance of such as unified codebase. MASLab will continue to evolve, tracking the latest developments in the field and incorporating advanced benchmarks, and welcome diverse contributions from the broader open-source community.

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

| Protocol | LLM-2step | LLM-xVerify | Rule-HF | Rule-DyLAN | Rule-Hendry. |
|----------|-----------|-------------|---------|------------|--------------|
| Accuracy | **98.59** | **98.35** | 41.65 | 65.65 | 27.29 |

Table 4: Accuracy comparisons of 5 different evaluation protocols by human's manual check. This measurement is based on MATH dataset. The two LLM-based evaluation protocols achieve significantly higher agreement with human evaluation. LLM-2step is based on two-time inference of Llama-3.3-70B-Instruct while LLM-xVerify is based on one-time inference of a 9B-sized LLM. Generally, LLM-xVerify achieves the best effectiveness-efficiency trade-off.

| Method | Level 1 | | Level 2 | | Level 3 | | All | |
|--------|---------|------|---------|------|---------|------|-----|------|
| | Acc | Cost | Acc | Cost | Acc | Cost | Acc | Cost |
| GPT-4.1-mini | | | | | | | | |
| Single | 16.98 | 663 | 16.28 | 353 | 0.0 | 1529 | 13.94 | 638 |
| SC | 22.64 | 4504 | 15.12 | 2412 | 0.0 | 8484 | 15.15 | 4041 |
| Debate | 24.53 | 4388 | 16.28 | 4870 | 7.69 | 12972 | 17.58 | 5992 |
| AgentVerse | 32.08 | 7174 | 15.12 | 7368 | 7.69 | 15753 | 19.39 | 8627 |
| OWL-Roleplaying | 35.85 | 51543 | 25.58 | 58881 | 11.54 | 107635 | 26.67 | 64206 |
| ReAct-MASLab | 33.96 | 19866 | 26.74 | 41768 | 11.54 | 55743 | 26.67 | 36935 |
| GPT-4.1 | | | | | | | | |
| Single | 24.53 | 394 | 16.28 | 470 | 3.85 | 1378 | 16.97 | 589 |
| SC | 20.75 | 3037 | 16.28 | 3362 | 11.54 | 11786 | 16.97 | 4585 |
| Debate | 32.08 | 4103 | 24.42 | 4339 | 11.54 | 11564 | 24.85 | 5402 |
| AgentVerse | 28.30 | 6876 | 18.60 | 5995 | 3.85 | 11034 | 19.39 | 7072 |
| OWL-Roleplaying | 43.40 | 48073 | 30.23 | 101827 | 26.92 | 101986 | 33.94 | 84586 |
| ReAct-MASLab | 56.60 | 18278 | 47.67 | 35636 | 19.23 | 43525 | 46.06 | 31303 |

Table 5: Comparisons of performance and cost on GAIA. The performance is evaluated by accuracy while the cost is evaluated by the number of costed text tokens per query.

## A    LLM Usage Disclosure

In our work, we used GPT-4 to improve readability and language fluency through polishing, and it was used solely during the writing process. We are solely responsible for the entire content of this publication, including any contributions generated by the LLM.

## B    Limitations

Despite being the most comprehensive codebase in LLM-based MAS, there are still methods that have not been incorporated yet. Secondly, despite that most of the benchmarks in this paper are commonly used in MAS literature, they are not specifically designed for the field of MAS. However, this is not a unique limitation of this paper. We will continually working on this codebase to support more methods and benchmarks. We also plan to design new benchmarks specifically for MAS in the future.

## C    Broader Impacts

This paper introduces a unified, comprehensive, and research-friendly codebase for the community of LLM-based MAS. This resource alleviates the burden of reproduction for researchers, enabling them to allocate more effort to innovative algorithm design. It fosters fair comparisons across studies, lowers the entry barrier for newcomers, and facilitates secondary development, thereby accelerating progress in the field.

While potential negative impacts of our approach mirror those associated with large language models—such as ethical concerns and risks of misuse—these issues are intrinsic to LLM usage in general and do not necessitate further elaboration here.

| Run | Optimization | | | Inference | |
| --- | --- | --- | --- | --- | --- |
| | Optimizer Cost | Executor Cost | Val Acc | Test Acc | Cost |
| MASLab | 0.58251$ | 19.05964$ | 54.52 | 65.20 | 1.489$ |
| Official | - | 19.52409$ | 53.27 | 65.06 | 2.231$ |

Table 6: Comparisons of our implementation of AFlow (Zhang et al., 2025b) and the official one. The optimizer is Claude-3.5-Sonnet while the executor is GPT-4o-mini. The official code does not record the optimizer cost. This table verifies the effectiveness of our re-implementation.

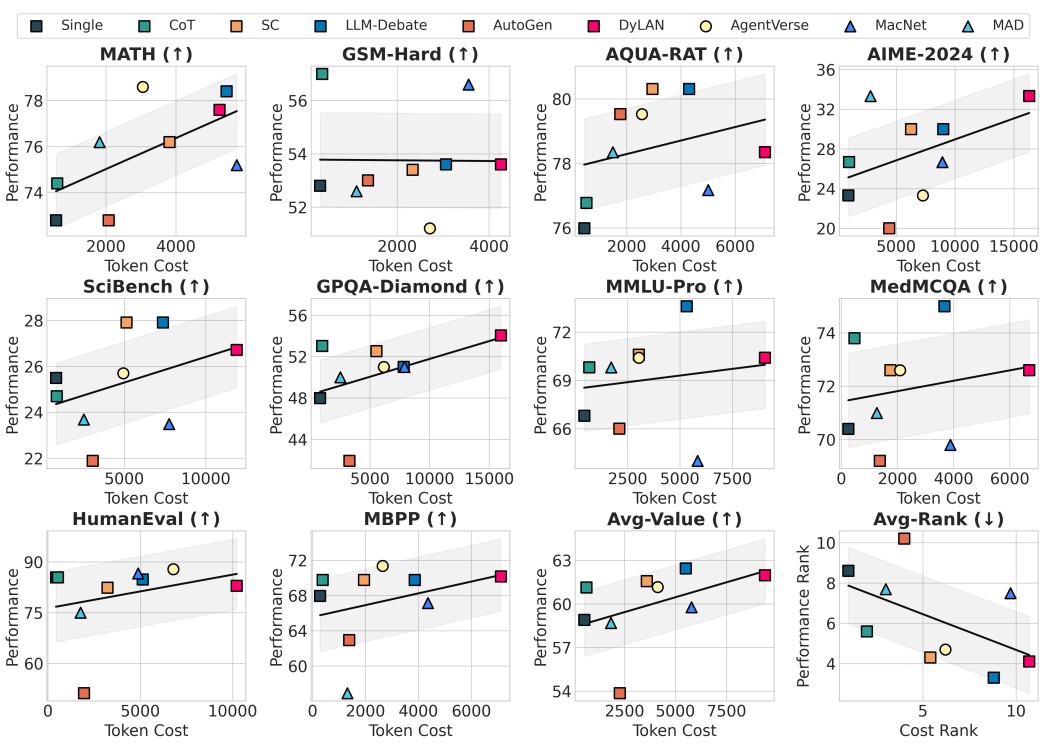

Figure 11: Examinations of trade-offs of performance and cost of nine MAS methods across 10 benchmarks.

# D  IMPLEMENTATION DETAILS

## D.1  COMPUTATIONAL RESOURCES

For open-source LLMs, we leverage the vLLM (Kwon et al., 2023) library launch LLM service. For 32B-, 70B-, and 72B-sized LLMs, we use 4 NVIDIA A100 GPUs; for 14B-sized LLMs, we use 2 NVIDIA A100 GPUs; for 7B-sized LLMs, we use 1 NVIDIA A100 GPU.

## D.2  GAIA

GAIA is a challenging benchmark for general AI assistants. In our experiments, we utilize the validation set of GAIA, which contains a total of 165 samples categorized into three levels of difficulty. It requires the MAS to engage in multi-turn collaboration to solve the tasks. Both the OWL-Roleplaying and React-MASLab methods are constrained to a maximum of 12 turns per task.

**Toolkits.** All methods share a common set of toolkits, including a web interaction tool, a document processing tool, a video analysis tool, an audio analysis tool, a code execution tool, an image analysis tool, a search tool, and an Excel tool. Several of these tools incorporate multimodal large language

models. Except for the audio analysis tool, all such tools utilize the same model version as the one configured in the main experimental pipeline. The web interaction tool employs the Playwright library to simulate browser behavior. However, we observe occasional instability during experimentation. To reduce both runtime and token consumption, we impose strict operational constraints: a 30,000 ms timeout for website navigation, a 20,000 ms timeout for page loading, and a hard cap of 10 web interaction turns per task. Tasks exceeding this limit are forcibly terminated. The document processing tool supports parsing a wide range of document formats. For web content extraction and parsing specifically, we employ an external tool called Firecrawl. The video analysis tool extracts 28 evenly spaced frames from each video and uses OpenAI's Whisper-1 model to transcribe the audio into text. These frames, along with the transcribed text, are jointly input into a vision-language model for multimodal analysis. The audio analysis tool processes audio files by encoding them in Base64 format and feeding them into the GPT-4o-mini-audio-preview model for analysis. The code execution tool operates by spawning a subprocess that simulates the writing and execution of Python code in a sandboxed environment. The search tool integrates multiple retrieval backends such as Google, DuckDuckGo, Wikipedia, and Archive.org, allowing agents to gather information from diverse sources.

**Memory.** We simplify the process of memory storage and retrieval for the model. To strike a balance between performance and token efficiency during memory retrieval, we impose a maximum limit of 51,200 tokens on the retrieved content. Similarly, we cap the maximum token length for model output at 12,800 tokens.

**Failure analysis.** Throughout the experiments, we log MAS outputs and failure cases. After the experiments, we select results from the OWL-Roleplaying method running on the GPT-4.1 model and perform a detailed categorization and statistical analysis of the errors encountered.

## E   RE-IMPLEMENTATION NOTES

### E.1   MAS FOR GENERAL TASKS

**AutoGen (Wu et al., 2024).** Based on the examples proposed in the paper of AutoGen (Wu et al., 2024) and the guidelines provided in its official documentation (`https://microsoft.github.io/autogen/0.2/`), we have developed a foundational workflow that embodies its conversational characteristics, tailored to solve basic text-level problems with code execution and memory retention.

**AgentVerse (Chen et al., 2024).** AgentVerse provides several dataset-specific versions including those for MGSM and HumanEval. we have replicated workflows corresponding to datasets such as HumanEval and MGSM, aligning with those presented in the original AgentVerse repository (`https://github.com/OpenBMB/AgentVerse`) and its paper. Additionally, we develop a general workflow capable of solving common problems.

**LLM-Debate (Du et al., 2024).** We notice that the official code in `https://github.com/composable-models/llm_multiagent_debate` is not readily executable and that the code relies on an string operation for extracting answers from responses, which frequently causes errors. Therefore, we slightly modify the code by making it bug-free and rely on LLM for aggregating final answers. This significantly enhance the performance of LLM-Debate as it no longer encounter errors during execution.

**GPTSwarm (Zhuge et al., 2024).** The official code of GPTSwarm `https://github.com/metauto-ai/GPTSwarm/tree/main/experiments` contains versions for MMLU, HumanEval, GAIA, Crosswords. We implement the version of HumanEval and MMLU, and based on the logic of MMLU, we develop a version for general problem-solving.

**DyLAN (Liu et al., 2024).** The official code in `https://github.com/SALT-NLP/DyLAN` uses a custom answer extraction function to return final mathematical results. To ensure fair comparison across evaluation protocols, we modify the return logic of the original code while preserving the task-specific initialization parameters as defined in the original implementation.

**Self-Refine (Madaan et al., 2024).** The official implementation in `https://github.com/madaan/self-refine` provides dataset-specific prompt examples. Following its logic for solving mathematical problems, we develop code for general problem-solving. Additionally, since the original

code's extraction logic for mathematical problems is not robust and often results in syntactically incorrect code, we redesign the extraction function to more effectively extract executable code from raw LLM responses.

**MacNet (Qian et al., 2025).** We simplify the structure of waiting.py in `https://github.com/OpenBMB/ChatDev/tree/macnet` when reproducing MacNet, but keep its functionality consistent, mainly in terms of high maintainability and memory safety. In addition to this, we develop a version for general cases according to their implementation for SRDD.

**Reflexion (Shinn et al., 2023).** For the method in `https://github.com/noahshinn/reflexion`, we implement the HumanEval and MBPP modes for programming tasks. Additionally, based on the logic of the programming tasks, we develop a version for general problem-solving.

**ADAS (Hu et al., 2025b).** We notice that the official code in `https://github.com/ShengranHu/ADAS` does not support flexible selection of execution models, which makes it difficult to evaluate the effect of the MAS module and to develop a heterogeneous MAS version. Therefore, we slightly modify the code to fix existing bugs and to allow users to specify both the meta LLM and the execution LLM during optimization, as well as choose the execution model during inference. We also set the temperature to zero and ensure that when using GPT-3.5 as the execution model (same as in the original repo), the output remains exactly the same. These improvements significantly enhance the compatibility and extensibility of ADAS.

**AFlow (Zhang et al., 2025b).** The official code in `https://github.com/FoundationAgents/MetaGPT/tree/main/examples/aflow` and `https://github.com/FoundationAgents/AFlow` is very complex and even buggy, we simplify the format and make sure that the core parts are fully aligned and bug free. In addition we use AsyncOpenAI when reproducing AFlow to speed up the optimization.

**MAV (Lifshitz et al., 2025).** We reproduce the MATH and MMLU versions of MAV and develope a general version based on the MATH version.

## E.2 MAS FOR CODING TASKS

**MetaGPT (Hong et al., 2024).** MetaGPT is an intricate system that presents a considerable challenge to analysis. Our research reveals that the efficacy of its communication infrastructure on the entire system is negligible, and its practical impact is confined to modest-scale projects. To facilitate comprehension, we streamline it into a linear framework, aligning it with the structure of the original paper. In fact, we find that the existing structure cannot be applied to datasets such as HumanEval and MBPP.

**ChatDev (Qian et al., 2024).** ChatDev primarily focuses on the domain of software development. By leveraging natural language processing techniques, ChatDev enables seamless automation of the entire software development lifecycle, including the generation of GUIs (graphical user interfaces). The complexity of the resulting software is closely tied to the specificity of user-defined requirements. Based on the official ChatDev paper ((Qian et al., 2024)) and its official repository (`https://github.com/OpenBMB/ChatDev`), we adapted a ChatDev workflow within the MAS-Lab framework tailored to SRDD (Software Requirement Description Dataset), aligning with the design principles and capabilities demonstrated in the original ChatDev system.

**MapCoder (Islam et al., 2024).** Our implementation follows the official codebase from `https://github.com/Md-Ashraful-Pramanik/MapCoder`, preserving its core methodology. However, we note that the original implementation uses a pre-processed version of the HumanEval dataset, which includes example test cases. To ensure a fair comparison across different methods, we do not use this pre-processed version. Instead, we augment the framework with a function that dynamically extracts test cases from the original HumanEval prompts. This modification does not affect MapCoder's core logic but ensures all baselines are evaluated under identical conditions.

**EvoMAC (Hu et al., 2025c).** We collaborate with the authors of EvoMAC, who provide their official implementation to be integrated into our framework. The method remains unchanged. Together with the authors, we release this joint implementation as part of our open-source framework, maintaining full transparency and reproducibility.

### E.3 MAS FOR MATHEMATICAL TASKS

**MACM (Lei et al., 2024).** MACM is an MAS method specialized in solving mathematical problems using the code interpreter tool to assist problem solving. Since their official code is specifically designed for the usage of OpenAI's Assistants interface, we follow the same LLM usage for this particular case. In the future, we plan to extend it to support OpenAI's chat mode.

### E.4 MAS FOR SCIENTIFIC TASKS

**MedAgents (Tang et al., 2024b).** The official code in `https://github.com/gersteinlab/MedAgents` supports multiple working modes. We fully reproduce all modes and set the default mode to match the original repository's default configuration, keeping all other external parameters consistent with the original defaults.

### E.5 MAS FOR TOOL-REQUIRED TASKS

**OWL-Roleplaying (Hu et al., 2025a).** OWL (`https://github.com/camel-ai/owl`) is a framework for multi-agent collaboration. This framework includes OWL-Roleplaying as a MAS method specifically designed for the GAIA benchmark (Mialon et al., 2024). This framework may introduce massive token consumption for each specific task/query. Considering the research-friendly nature of the our MASLab framework, several trade-offs and simplifications are made during the adaptation of this method to MASLab, with a focus on enhancing code readability and reducing computational costs. Overall, the main process of OWL is maintained during the adaptation while we limit the maximum retrying times considering economy. For example, we set stricter limitations on the use of the web tool to mitigate the substantial token costs associated with frequent web interactions.

**ReAct-MASLab (Yao et al., 2023).** Building upon the toolkits from OWL, we propose a method ReAct-MASLab inspired by the ReAct (Yao et al., 2023) method. This method achieves better performance with lower cost compared to OWL-Roleplaying.

