# OpenReview forum: "MASLab: A Unified and Comprehensive Codebase for LLM-based Multi-Agent Systems"
_ICLR.cc/2026/Conference — ICLR 2026 Conference Withdrawn Submission_

### Official Review · Reviewer_mwB4 · 2025-10-16

**Soundness:** 3
**Presentation:** 3
**Contribution:** 2
**Rating:** 4
**Confidence:** 4

**Summary:**

This paper presents MASLab, a unified codebase for multi-LLM systems.
MASLab integrates 20+ methods and 10 benchmarks, with the goal of removing re-implementation efforts, the possibility of unfair comparisons, and high entry bars for new researchers.
The authors conducted extensive empirical studies to analyze the performance, cost (in tokens) and scaling properties.

I think the paper is solving an important problem in multi-LLM research, but I also have 2 concerns.
I'm wiling to raise my score if they can be addressed.

**Strengths:**

* The primary strength of this paper is the codebase, it allows future researchers to conduct experiments in a unified and easier way.
* The empirical studies presented in the paper are extensive and comprehensive.

**Weaknesses:**

* The codebase is not available in this submission, making it hard to evaluate its easiness of adoption. Since the major contribution of this paper (i.e., a unified codebase) is more engineering than novel research, the lack of the source code makes the impact less convincing.
* Although the authors claimed that step-by-step output verification was conducted to ensure a validated implementation, quantitative evidence is insufficient (e.g., Table 6 shows the comparison in AFlow, but the rest methods are unknown). While providing source code can partly address this problem, I would like to see quantitative results reproducing 50%+ of the methods. For a paper whose primary contribution is a reliable library, this is a significant cap.
* (Minor) Typos, e.g., line 313 "able 2".

**Questions:**

Can the authors provide source code and reproduce the results for more methods? See weakness 1 and 2.

---

### Official Review · Reviewer_cgcz · 2025-10-20

**Soundness:** 3
**Presentation:** 3
**Contribution:** 1
**Rating:** 2
**Confidence:** 4

**Summary:**

The authors propose a codebase for LLM-based multi-agent systems, called MASLab. This codebase includes twenty existing agentic solutions, with unified data preprocessing, resource, configuration, and evaluation. The authors present plenty of experimental results on different benchmarks to demonstrate the results of different solutions implemented in MASLab.

**Strengths:**

1. The authors survey the most popular agentic solutions comprehensively, in terms of the features of agentic solutions and their applicable areas.
2. The authors conduct a wide range of agentic benchmarks to compare the adapted versions of the agentic solutions in their MASLab.
3. The paper presents the key results in a way that readers can understand with little effort.

**Weaknesses:**

1. A key concern of this paper is its limited scientific contribution to the community. This reads essentially as a benchmark paper that evaluates various of agentic solutions. However, this paper also introduces some additional modifications to those solutions/frameworks, which may make the overall performance attribution more difficult. Namely, it is unclear whether the "unifying" operations and adaptations introduce the performance degradation/improvement compared to their original implementation.
2. Another factor that blur the paper's contribution is that although it claims that a significant effort is spent in building such a unified codebase, the paper does not explicitly demonstrate the benefit of the system, such as how it becomes easier for beginners, what the benefits of using the new system are, and how general of the high-level system structure can be.
3. Some experiment designs and their conclusions seem too casual or even meaningless, such as "Equipping agents with tools improves MAS performance" for the GAIA benchmark.
4. "MASLab-ReAct" was never introduced but appears in experiments.

**Questions:**

1. Why can different evaluation protocols introduce such differences as shown in Figure 3? How does it relate to the capability of LLMs used in judging and the backbone of the agent?
2. How can the paper guarantee that the unifying operation on the input and other components does not introduce unnecessary performance degradation? How to ensure the contexts fed to LLMs in the unified ones have the same effect as the original solutions or better than those?
3. In the error analysis, how can one tell whether the error is because of "incorrect final answers" or "errors in tool usage"? Or, what scenarios exactly are included in "incorrect final answers"?

---

### Official Review · Reviewer_MHpj · 2025-11-01

**Soundness:** 2
**Presentation:** 3
**Contribution:** 3
**Rating:** 6
**Confidence:** 3

**Summary:**

The paper introduces MASLab, a unified and comprehensive codebase for LLM-based multi-agent systems (MAS). MASLab addresses key challenges in the field, including redundant implementations, inconsistent evaluation protocols, and high entry barriers for researchers. It integrates over 20 state-of-the-art MAS methods across various domains (e.g., general tasks, coding, mathematics, science) and provides a standardized evaluation framework to ensure fair and reproducible comparisons.

**Strengths:**

1. The paper introduces MASLab, which consolidates over 20 state-of-the-art MAS methods across multiple domains (e.g., general tasks, coding, mathematics, science). This integration reduces redundant implementation efforts
2. MASLab addresses a critical gap in the field by offering a standardized evaluation pipeline. It ensures consistent input preprocessing, configuration alignment, and evaluation protocols, which are essential for fair and reproducible comparisons.
3. By abstracting each MAS method into a streamlined Python class, MASLab makes it easier for researchers to understand, extend, and innovate upon existing approaches. This design also facilitates secondary development, making it accessible for both newcomers and experienced researchers.

**Weaknesses:**

1. While the paper evaluates MAS methods across 10+ benchmarks, many of these benchmarks are not specifically designed for MAS.
2. In the evaluation system, the default xVerify method is supervised. Does this imply that users must train new evaluation models when extending to new assessment tasks?
3. The work lacks theoretical or methodological innovation. Its focus is primarily on engineering and software development. Although some interesting phenomena were observed during the evaluation of different MASs, there was no in-depth analysis or exploration of the underlying causes or principles of these phenomena. For instance, the paper mentions the significant impact of evaluation protocols on the final results and analyzes failure cases of MAS systems; however, it does not provide detailed analyses of these findings and instead emphasizes engineering solutions to these issues.

**Questions:**

1. Can some recent benchmarks specifically designed for MAS better highlight the relevance of MASlab?
2. What are the scalability and computational costs of the xVerify method? Does it support users in freely extending to arbitrary evaluation tasks and datasets?

---

### Official Review · Reviewer_qex7 · 2025-11-01

**Soundness:** 3
**Presentation:** 3
**Contribution:** 2
**Rating:** 4
**Confidence:** 4

**Summary:**

This paper presents a new LLM-based multi-agent system (MAS) framework that serves as a unified platform for implementing, running, and comparing diverse agent methods. The proposed codebase supports more than 20 existing approaches and provides standardized environments for benchmarking and evaluation. It is designed to be easily extensible, allowing integration of new methods. Experiments conducted on over ten benchmark suites demonstrate the usability and versatility of the framework.

**Strengths:**

This paper proposes a codebase (framework) that combines two capabilities: easy implementation and unified execution, as well as wide-ranging, standardized benchmarking. This identifies a real gap in prior work, as existing methods typically focus on only one of these aspects. The framework also includes a wide range of recent methods and benchmarks, which strengthens the contribution.

The experiments show broad coverage and a rigorous, thorough evaluation. This is a good indication of a versatile benchmark framework proposed in the paper.

**Weaknesses:**

Novelty: While it is reasonable to claim that this work conceptually fills the gap between easily extensible multi-agent frameworks and broad benchmark coverage, prior works are not entirely lacking in this direction. For example, CRAB (https://github.com/camel-ai/crab
, https://arxiv.org/abs/2407.01511) supports a much larger number of tasks (120) while maintaining a modular shared codebase. AgentBoard (https://openreview.net/forum?id=4S8agvKjle), while focusing mainly on unified environments and consistent metrics for comparison, also supports some degree of agent customization. In addition, AgentVerse (https://github.com/OpenBMB/AgentVerse) provides extensibility for agent methods, and CRAB allows extensibility for environments, although not simultaneously. Since this paper does both, such overlap between prior works and this study does not fully undermine the innovative aspect, but the paper would benefit from a more detailed discussion of these prior efforts and a clearer explanation of what specifically distinguishes it from them.

Claimed versatility: As a framework and benchmark paper, the usability aspect (especially the claimed contribution in ease of extension) cannot be fully assessed without access to the code. Although the manuscript provides sufficient experimental details, they are not fully convincing without demonstrating actual usability through a released implementation. Even without providing an anonymous repository (understandable if that is not feasible during rebuttal), some code excerpts or minimal skeletons for agents and tasks would still be helpful in demonstrating the framework’s design.

Experiments: Given that the framework aims to be versatile, it should support and test a wider range of LLM models in addition to the already broad coverage of methods and tasks. Currently, it seems that the choice of underlying LLMs (a subset of the Llama, GPT, and Qwen families for each task) is task-specific without clear justification. Ideally, the choice of LLM models should be orthogonal to tasks and environments unless there is incompatibility, so an explanation or a more systematic model-swapping study would be expected.

Overall, these issues are not fundamental flaws but rather reflect unclear novelty and incomplete validation of what appears to be a solid engineering contribution. Currently, this work is slightly below threshold due to unclear novelty, but the score could be increased if these issues are addressed and the unclear points clarified.

**Questions:**

(See the unclarities in Weakness)

---

### Note · Authors · 2026-01-09

I have read and agree with the venue's withdrawal policy on behalf of myself and my co-authors.